# Functional analysis of promoter element 2 within the viral polymerase gene of an emerging paramyxovirus, Sosuga virus

Lipi Akter,[1] Junna Kawasaki,[2] Tofazzal Md. Rakib,[3] Takashi Okura,[4] Fumihiro Kato,[5] Shohei Kojima,[6] Kosuke Oda,[7] Yusuke Matsumoto[1]

**ABSTRACT** Paramyxovirus genomes carry bipartite promoters at the 3′ ends of both their genome and antigenome, thereby initiating RNA synthesis, which requires the viral polymerase to recognize two elements: the primary promoter element 1 (PE1) and the secondary promoter element 2 (PE2). We have previously shown that the antigenomic PE2 (agPE2) in many viruses in the *Rubulavirinae* subfamily is located within the coding region of the viral RNA polymerase L gene. Sosuga virus (SOSV), belonging to the *Rubulavirinae* subfamily, is highly pathogenic to humans, thus necessitating high-level containment facilities for infectious virus research. The use of a minigenome system permits studies of viral RNA synthesis at lower biosafety levels. Because minigenomes of negative-strand RNA viruses generally comprise only the untranslated regions, agPE2 within the L coding region—such as those found in *Rubulavirinae* like SOSV—is typically omitted. However, generating an SOSV minigenome that retains agPE2 led to a pronounced increase in activity, enabling a detailed examination of the role of agPE2 in SOSV replication. In many *Rubulavirinae*, the agPE2 not only acts as a promoter but also encodes part of the L protein, resulting in a distinct motif at the C-terminus of the L protein. We have further shown that this motif is preserved even in *Rubulavirinae* that no longer contain the agPE2 within the L gene.

**IMPORTANCE** Paramyxoviruses are classified into three major subfamilies: *Orthoparamyxovirinae*, *Avulavirinae*, and *Rubulavirinae*. All paramyxovirus genomes and antigenomes possess bipartite promoters, comprising two elements: promoter element 1 (PE1) at the 3′ end and promoter element 2 (PE2) located internally. We previously revealed that, in many *Rubulavirinae*, the antigenomic PE2 lies within the coding region of the viral RNA polymerase L gene. In this study, we used Sosuga virus, a member of the *Rubulavirinae* subfamily, to elucidate the role of antigenomic PE2 in viral replication. Because the PE2 region encodes part of the L protein, its presence leads to a distinctive motif at the C-terminus of L protein. Notably, this motif is conserved in all *Rubulavirinae*, including those that do not harbor the antigenomic PE2 within their L gene, indicating its importance in viral propagation.

**KEYWORDS** negative-strand RNA virus, paramyxovirus, promoters, RNA polymerases, viral replication

Paramyxoviruses are single-stranded negative-sense RNA viruses that belong to the order *Mononegavirales*, which includes many important pathogens of animals and humans (1). The family *Paramyxoviridae* is mainly divided into the following subfamilies: *Orthoparamyxovirinae* (measles virus, Nipah virus, and Sendai virus), *Avulavirinae* (Newcastle disease virus), and *Rubulavirinae* (parainfluenza virus type 5 [PIV5], human parainfluenza virus type 2 [hPIV2], and mumps virus [MuV]). The negative-strand RNA genome of paramyxoviruses typically encodes about six genes (e.g., hPIV2; 3′-NP-P/

Address correspondence to Yusuke Matsumoto, ymatsu@vet.kagoshima-u.ac.jp.

Yusuke Matsumoto receives compensation from Denka Co., Ltd. The other authors declare no competing interests.

See the funding table on p. 14.

V-M-F-HN-L-5′), which are completely encapsidated by the viral nucleoprotein (NP) to form a helical nucleocapsid. This nucleocapsid serves as a template for the viral RNA-dependent RNA polymerase (RdRp) complex, which consists of the large protein (L) and the phosphoprotein (P) cofactor (2).

Viruses belonging to the order *Mononegavirales* use the 3′ terminus of their genome as a replication promoter (promoter element 1, PE1). In addition, viruses in the family *Paramyxoviridae* possess a bipartite promoter system that requires a secondary promoter element (PE2) located within the internal region of the genome (Fig. 1A) (3–6). The PE1/2 located at the 3′ end of the negative-strand genome is referred to as the genomic PE and functions in the production of the positive-strand antigenome. Conversely, the PE in the positive-strand antigenome is termed the antigenomic PE (agPE) and is essential for the production of the negative-strand genome. Furthermore, paramyxoviruses follow the "Rule of Six," which requires that the nucleotide (nt) numbers in the genome should be a multiple of six for efficient viral replication (3, 7). The genomic RNA is coated by NP monomers, with each monomer binding precisely to 6 nts (8–10).

Understanding the genome structure and replication mechanisms of paramyxoviruses is crucial for elucidating how these viruses propagate and exert their pathogenicity. Research on highly pathogenic viruses plays a significant role in developing new therapeutic and preventive strategies. Sosuga virus (SOSV) belongs to the genus *Pararubulavirus* in the subfamily *Rubulavirinae* (11–14). SOSV was first identified in clinical samples from a wildlife biologist who became infected while collecting bat specimens in South Sudan and Uganda in 2012. The patient exhibited symptoms including fever, generalized myalgia, arthralgia, neck stiffness, and a sore throat. Due to its pathogenicity, experiments using infectious SOSV must be conducted in high-biocontainment facilities at biosafety level (BSL)-4 (or in some cases, BSL-3). As a result, research on SOSV using the authentic virus is restricted, creating a need for genome replication modeling systems that allow for the safe study of the viral life cycle at lower biosafety levels. One such model is the minigenome system, which can be utilized in a BSL-2 environment. The development of an experimental system that efficiently measures the function of RdRp in highly pathogenic paramyxoviruses is required for a detailed study of the molecular basis of genome replication.

## RESULTS

### An improved SOSV minigenome system reveals the importance of agPE2 for replication

The minigenome system is designed to indirectly quantify viral gene expression by creating a construct in which a reporter gene is flanked by the 3′ and 5′ untranslated regions (UTRs) of the viral genome. When introduced into cultured cells along with the viral polymerase complex, this system enables the assessment of viral gene replication. In the case of SOSV, a minigenome system using Zoanthus green fluorescent protein (ZsGreen1) as a reporter gene has been previously reported (12). In this study, we developed a new system in which the reporter gene of the SOSV minigenome system was replaced with the easily quantifiable Nano Luciferase (Nluc). During the process for minigenome construction, the initiation codon of the first gene NP and the stop codon of the last gene L are replaced with the corresponding codons from ZsGreen1 or Nluc, and the total nucleotide count is adjusted to ensure it remains a multiple of six. The ZsGreen1 version of minigenome created by the process above was reported previously (12). We prepared a Nluc-expressing SOSV minigenome by the method above, termed as the minigenome-1 (Fig. 1C). The minigenome-1 expressing plasmid was transfected into baby hamster kidney (BHK) cells constitutively expressing T7 RNA polymerase (BHK/T7-9 cells) together with the SOSV NP, P, and L expressing helper plasmids, and Nluc expression was examined 48 h post-transfection. The minigenome-1 system showed only approximately five times the activity of the negative control (no NP expression plasmid) (Fig. 1C; minigenome-1). We have previously shown that the agPE2 of *Rubulavirinae* is located within the L open reading frame (ORF), termed PE2 In-ORF, with the exception of

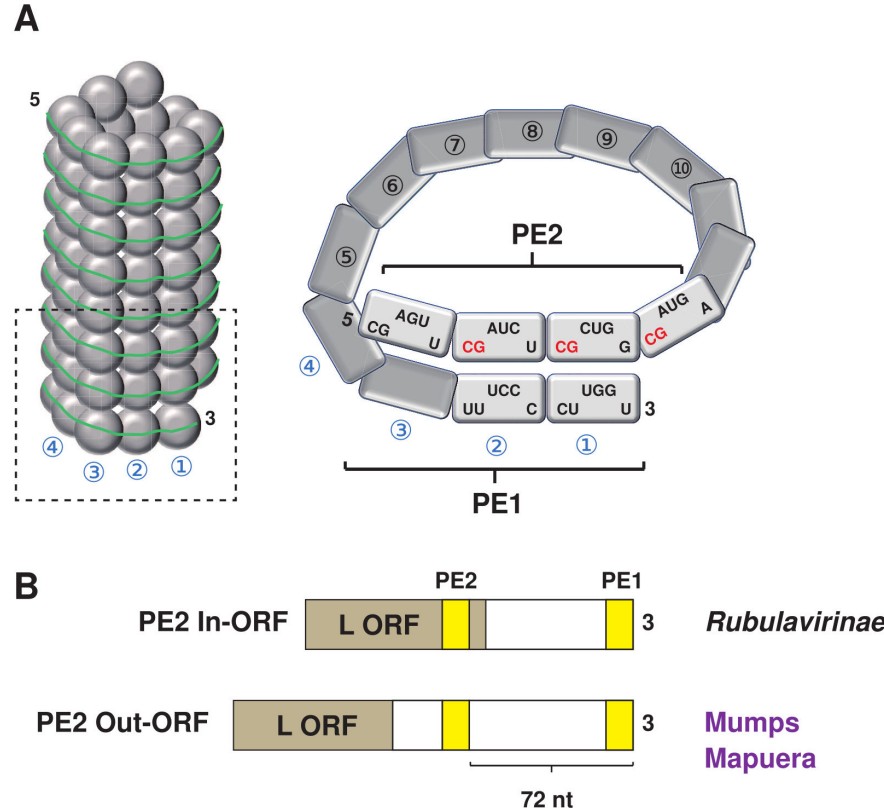

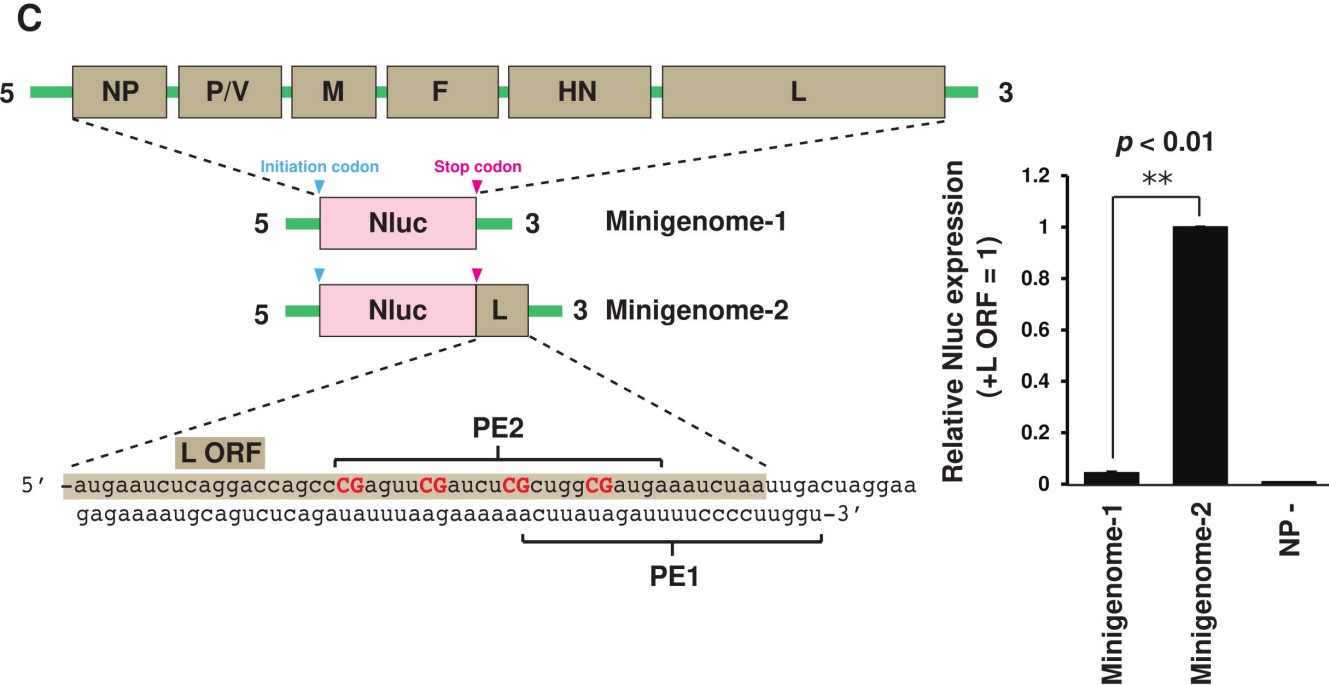

FIG 1 Establishment of Sosuga virus (SOSV) minigenome assay. (A) Helical structure of SOSV nucleocapsid. Promoter element 1 (PE1) and PE2 are shown. (B) Viruses in the *Rubulavirinae* subfamily are categorized into PE2 In-ORF and Out-ORF types. (C) Schematic diagram of SOSV minigenome construction. All viral genes in the genome are exchanged to be Nluc. Because the SOSV genome contains the PE2 within the L gene, two minigenomes with or without PE2 are created. The relative Nluc expression using the SOSV minigenome assay is shown. Data represent means and standard deviations from three different experiments. NP− indicates the values using an empty plasmid instead of SOSV NP.

MuV and Mapuera virus, termed PE2 Out-ORF (Fig. 1B) (15). This SOSV minigenome-1 does not retain the correct agPE2 (12). Therefore, we prepared the minigenome-2 by incorporating part of the nucleotides corresponding to the L ORF after the stop codon of the Nluc gene (Fig. 1C; minigenome-2). The minigenome-2 system showed 21 times the activity of the minigenome-1 system and 106 times the activity of the negative control NP−. This shows that agPE2 in the L ORF is essential for the replication activity of SOSV.

## Importance of CG repeats in agPE2 for SOSV minigenome replication

The SOSV minigenome-2 system enables us to study the significance of CG repeats in viral replication. Although the three repeated CGnnnn in hexamers 13, 14, and 15 were shown to be important for PIV5 (*Rubulavirinae*) genome replication, SOSV agPE2 possesses four consecutive CGnnnn repeats. The activity of the minigenome was assessed by systematically replacing the CGs in the four CGnnnn repeats of SOSV agPE2 with AU pairs, 1–4 at a time (Fig. 2). When a single pair of CGs was substituted, the activity dropped to 15% upon replacing hex14 with AU; however, when hex13, hex15, and hex16 were substituted, the activity remained stable at 50%–60% (Fig. 2; mCG3 to mCG6). With two pairs of CG substitutions, it was observed that replacing hex13 and hex16 or hex15 and hex16 had a relatively minimal impact on activity (Fig. 2; mCG34 to mCG56). When three and four pairs were replaced, all substitutions exhibited a significant negative effect (Fig. 2; mCG345 to mCG3456). These results indicate that the CG in hex14 is critical for the SOSV promoter. The relatively high activity observed in the two pairs of mutations involving hex16, when hex14 was omitted (mCG36 and mCG56), suggests that the importance of hex16 may be somewhat lower than that of the other repeats. It is suggested that the three or four CG repeats within agPE2 in SOSV are not equally important, but rather, their significance for genome replication varies.

## The relationship between the positions of PE1 and PE2 and their function as replication promoters

The paramyxoviral genome with bipartite promoters is recognized by viral RdRp as a template only when NP-encapsidated PE1 and PE2 adopt a configuration at the 3′ end of the helical nucleocapsid, as depicted in Fig. 3A (3, 4). During genome (or antigenome) replication, the NP encapsidates the newly synthesized genome/antigenome incrementally by 6 nts from the 5′ end (16). If the total number of nucleotides in the genome/antigenome is not a multiple of six, the relative positions of PE1 and PE2 become misaligned. In this case, it is considered that the viral RdRp cannot recognize the correct promoters. This is thought to be a mechanism for RdRp to recognize that the template is a multiple of six. We added 1–5 nts to the front and back of agPE2 in the SOSV minigenome-2, assessing their replication activity when they were not multiples of six (Fig. 3A). In the original state, agPE1 and agPE2 are correctly aligned: 5′-CGcugg-3′ in hex14 is aligned with 5′-cuuggu-3′ in hex1, and 5′-CGaucu-3′ in hex15 is aligned with 5′-uuuccc-3′ in hex2 (Fig. 3A; original). The method of adding 1 to 5 nucleotides to the front of agPE2 produces a 6n + 1 to 6n + 5 minigenome without changing the juxtaposition of nucleotide sequences in [hex14 and hex1] and [hex15 and hex2] (Fig. 3; front +1 to +5). In this case, the SOSV minigenome activity was slightly observed, and in particular, the front +1, +3, +4, and +5 minigenomes showed about 20% of the original activity (Fig. 3B; front). The method of adding 1 to 5 nucleotides to the back of agPE2 changes the order of [hex14 and hex1] and [hex15 and hex2], but the natural nucleotide sequence within the hex13 to hex16 NPs is not changed (Fig. 3; back +1 to +5). In all cases of these mutations, the SOSV minigenome activity resulted in a remarkable loss of activity (Fig. 3B; back). These results suggest that it is not the specific placement of the CG motif within NP that is essential for promoter activity, but rather the preservation of the parallel arrangement between PE1 and PE2—even if the precise positioning of the CG motif within NP is lost.

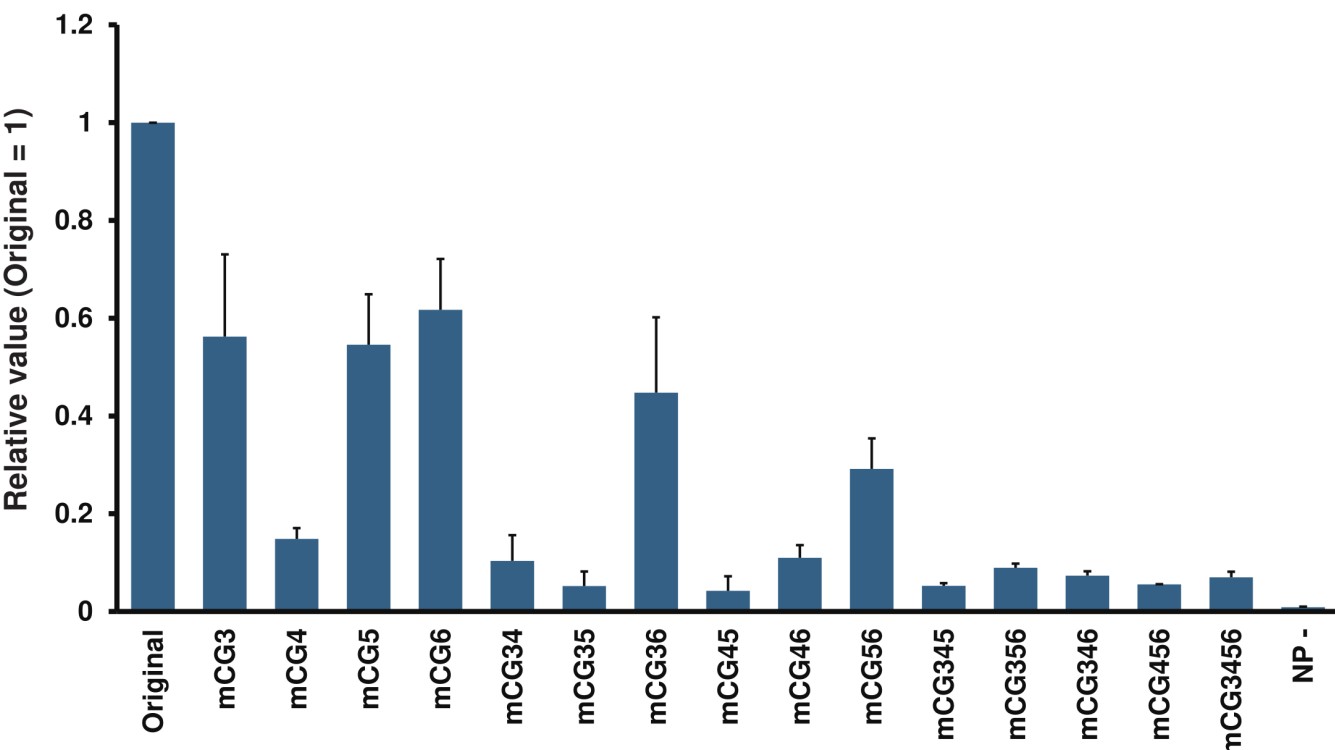

FIG 2  Importance of CG repeats in PE2 for SOSV genome replication. Mutations in the PE2 sequences are shown in the left panel. The relative Nluc expressions using the SOSV minigenome assay system are shown in the right panel. The relative value when the value using an original minigenome is 1. Data represent means and standard deviations from three different experiments. NP– indicates the value using an empty plasmid instead of SOSV NP.

## Comprehensive analysis of agPE2 of the subfamily *Rubulavirinae*

To gain more insight into agPE2 function of all other *Rubulavirinae*, we comprehensively compared the conservation of nucleotides among all *Rubulavirinae* agPE2 in sequences in the National Center for Biotechnology Information (NCBI) refseq database. The agPE2 region (73–96 nts from the 3′ terminus of the antigenome) was extracted from all

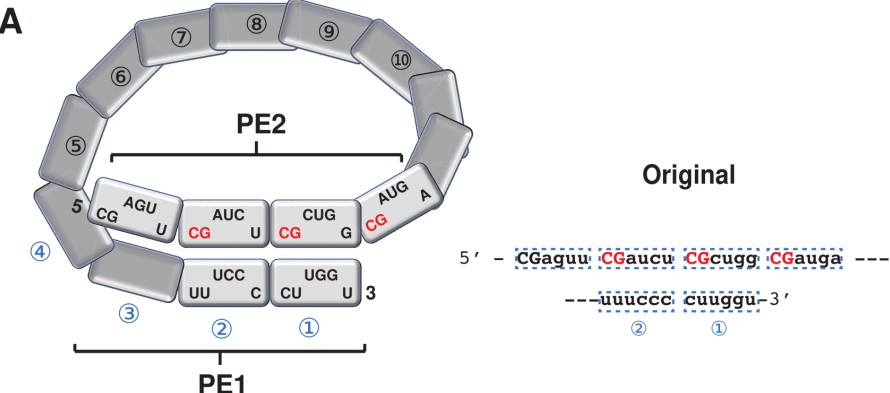

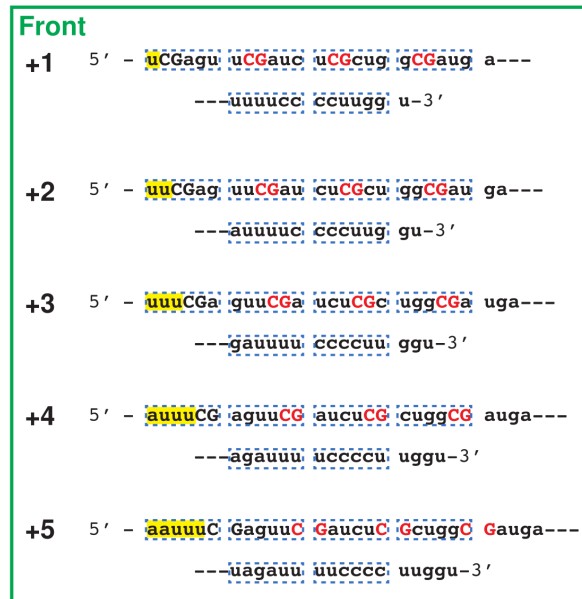

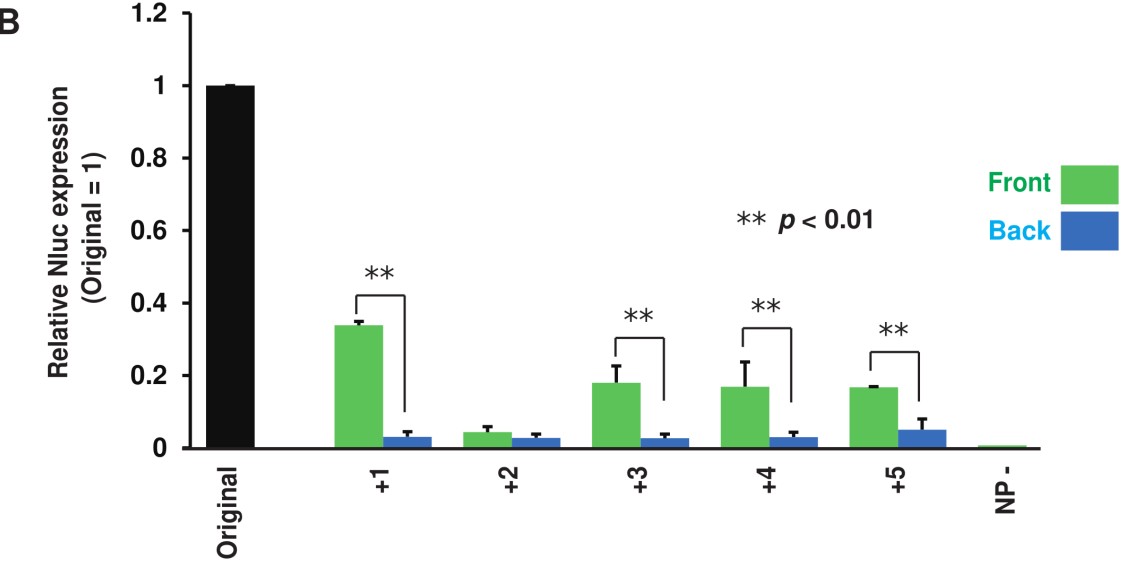

**FIG 3** The replication of SOSV minigenome constructed by nucleotides not a multiple of six. (A) Effect of nucleotide addition in the 5′ or 3′ position of agPE2 for minigenome expression. One to five nucleotide additions (+1 to +5) at the indicated position and their effect for the shift between PE1 and PE2 are shown. (B) The relative value when the value using an original minigenome is 1. Data represent means and standard deviations from three different experiments.

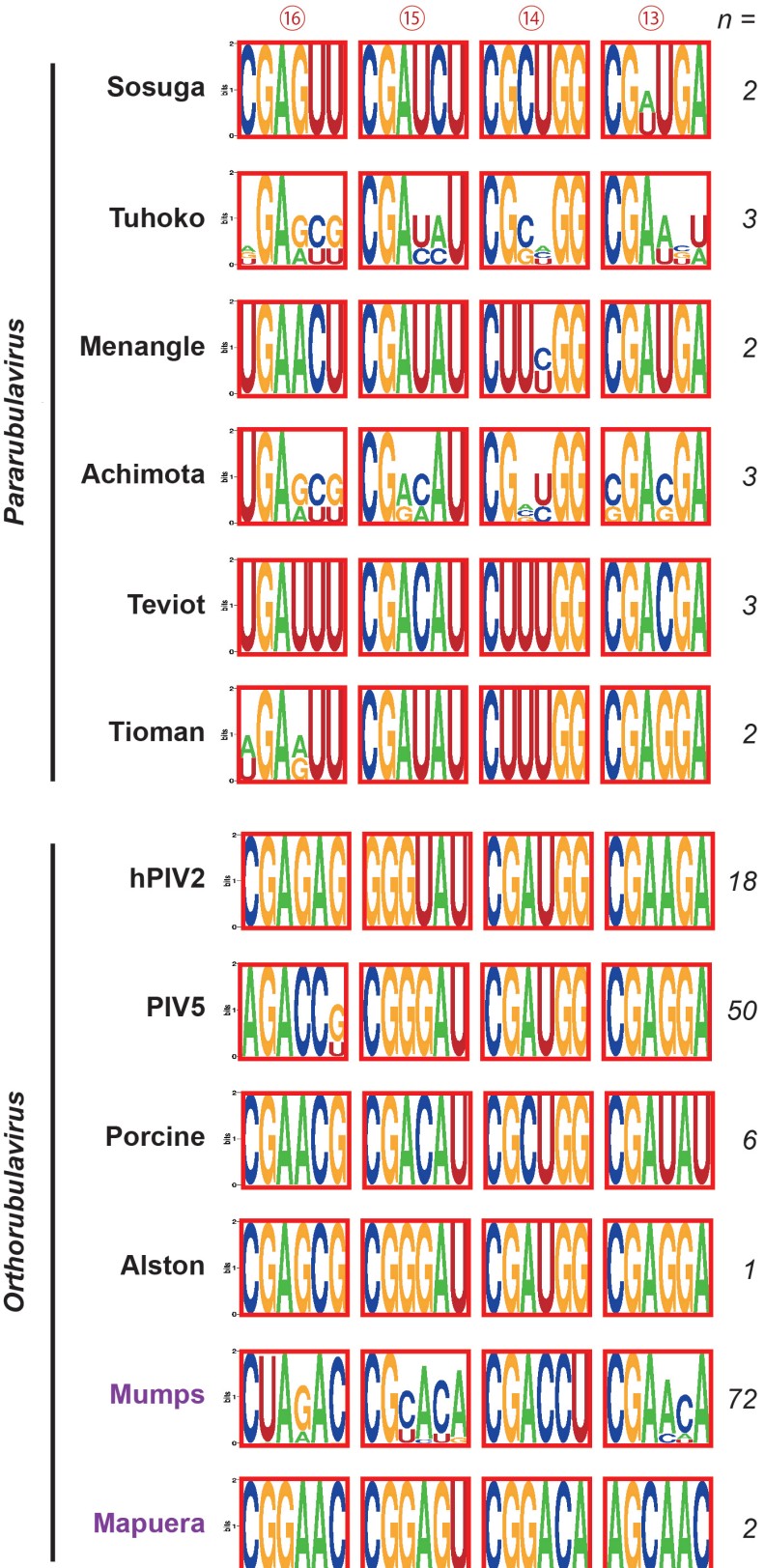

**FIG 4** Comparison of the agPE2 sequence of viruses in the subfamily *Rubulavirinae*. The agPE2 sequence of viruses in the genera *Pararubulavirus* and *Orthorubulavirus* are shown. Numbers in the circles indicate the numbers of the NP hexamer from the 3′ terminus. hPIV, human parainfluenza virus; Porcine, Porcine rubulavirus.

sequences, and their conservation among virus species was analyzed using the WebLogo sequence logo generator (https://weblogo.berkeley.edu/logo.cgi) (Fig. 4). There are a total of 12 species of *Rubulavirinae* whose sequences have been accurately registered in NCBI refseq, including six each from the genera *Pararubulavirus* (SOSV, Tuhoko, Menangle, Achimota, Teviot, and Tioman viruses) and *Orthorubulavirus* (hPIV2, PIV5, Porcine rubulavirus, Alston, mumps, and Mapuera viruses). Notably, SOSV, Porcine rubulavirus, and Alston virus exhibit four consecutive CGnnnn repeats within the hexamer numbers 13 to 16 from the 3′ terminal (namely hex13 to hex16) (Fig. 4). In the genus *Pararubulavirus*, Tuhoko and Achimota viruses, as well as in the genus *Orthorubulavirus*, PIV5 and MuV, the hex13, hex14, and hex15 exhibit three consecutive CGnnnn repeats. Several other patterns emerged, including hPIV2, which appears to have four consecutive CGnnnn repeats interspersed with a single GGnnnn, and Mapuera virus, characterized by three CGnnnn repeats in hexamers 14, 15, and 16. Overall, our findings indicate the conservation of CGnnnn repeats of three or more in the agPE2. We observed no clear distinctive features in the agPE2 sequences between PE2 In-ORF viruses and PE2 Out-ORF viruses (MuV and Mapuera virus).

## Amino acid sequences of L protein at the agPE2 region

The agPE2 region of In-ORF *Rubulavirinae* serves dual roles as a promoter element and as a codon for amino acids in the L protein. We investigated the specific amino acids encoded by the agPE2 region across all *Rubulavirinae* L proteins including In-ORF and Out-ORF viruses (Fig. 5A). This region is situated at the C-terminus of the L protein and is characterized by 3–4 CGnnnn repeats, which restrict codon usage. Consequently, as illustrated in Fig. 5B, these amino acid sequences exhibit the D/G-I/L-x-G-B motif (where B denotes D or E). Notably, MuV and Mapuera virus, which are PE2 Out-ORF viruses, lack agPE2 in the L ORF, but the amino acid sequences at the same position of the C-terminus of the L protein have the same D/G-I/L-x-G-B motif as In-ORF *Rubulavirinae*.

We constructed a predicted structural model of the SOSV L protein to visualize the position of the $D_{2265}LAGD_{2269}$ motif within the SOSV L context (Fig. 5C). The structure of the C-terminal loop containing the $D_{2265}LAGD_{2269}$ motif is relatively less reliable, perhaps due to protruding into the solvent region. In the model, this $D_{2265}LAGD_{2269}$ motif was bound to the concave on the molecular surface, which is surrounded by methyltransferase (MTase) domain, C-terminal domain, and connecting domain.

To assess the functionality of the $D_{2265}LAGD_{2269}$ motif of SOSV L protein, we replaced it with $A_{2265}AAAA_{2269}$ and used it as a helper plasmid for the SOSV minigenome-2 system (Fig. 5D). In this experiment, we used an original SOSV minigenome-2 (no mutation in its agPE2) as a template and transfected it with helper plasmids including SOSV NP, P, and L (normal or mutant). However, the $A_{2265}AAAA_{2269}$ mutation did not affect polymerase activity (Fig. 5D). Therefore, it appears that the SOSV L $D_{2265}LAGD_{2269}$ motif may not play a significant role in polymerase function, at least in the SOSV minigenome-2 system.

## DISCUSSION

For studying the genome replication mechanism of highly pathogenic negative-strand RNA viruses such as Ebola, Marburg, and Nipah viruses, the minigenome systems have been used (17–20). In the process of developing a minigenome system for pathogenic SOSV using Nluc as a reporter, we have shown that the minigenome activity increased by more than 20 times simply depending on the presence of agPE2 within the L ORF (Fig. 1C). Due to this high minigenome activity, we were able to analyze the properties of agPE2 for In-ORF type *Rubulavirinae*.

The paramyxovirus promoter operates only when NP-encapsidated PE1 and PE2 at the 3′ terminus of the helical nucleocapsid are properly aligned, as shown in Fig. 1A (3, 4, 21). In PIV5 and hPIV2, the repeated CG sequence within hexamers 13, 14, and 15 plays a crucial role in the agPE2 region (21–23). It was thought the specific positioning of the 5′-CG-3′ sequence within NP monomers (with C as the first and G as the second in

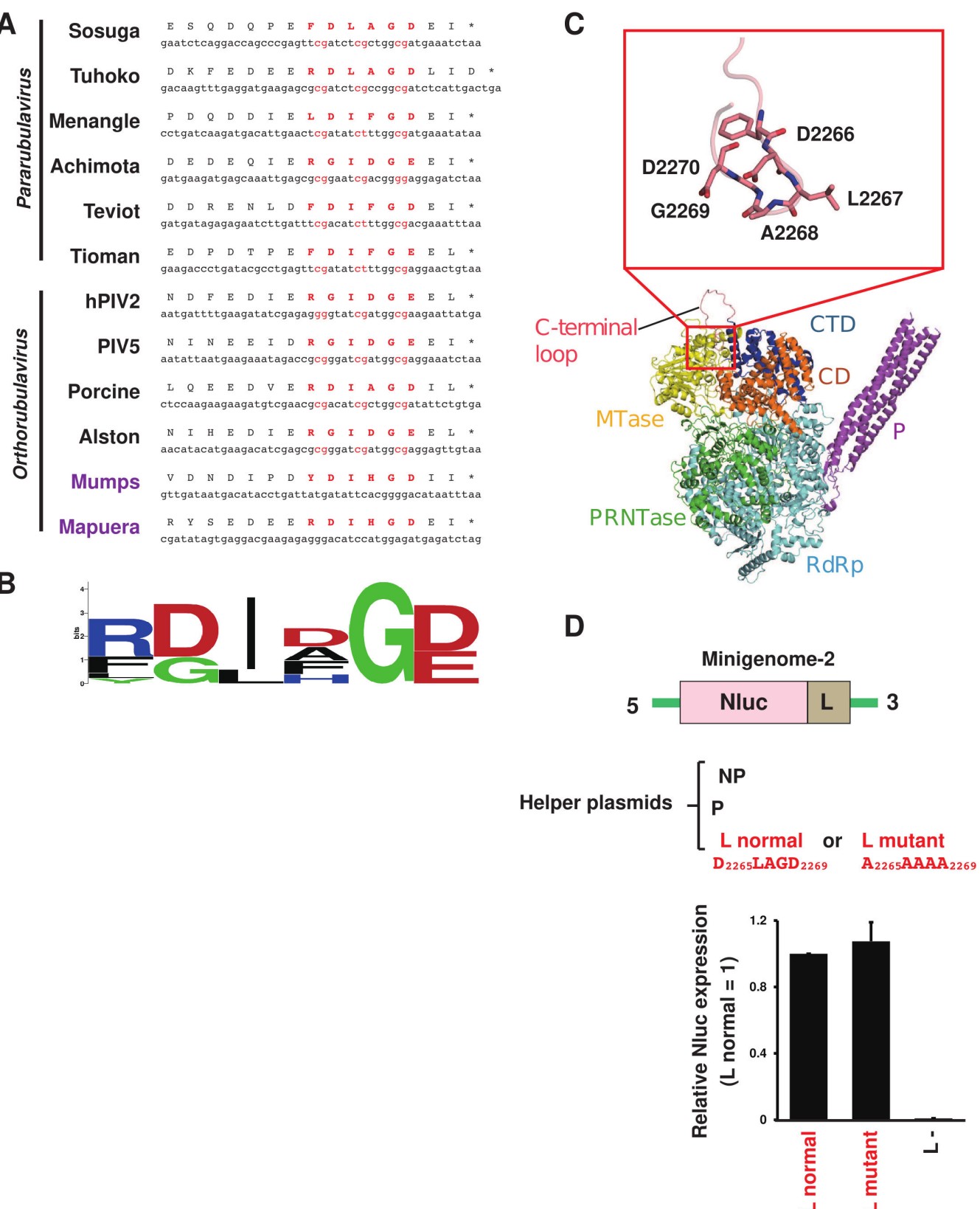

**FIG 5** Amino acid sequences of PE2 encoding region in the RNA-dependent RNA polymerase gene. (A) Comparison of nucleotide and amino acid sequences of PE2 of viruses in the genera *Pararubulavirus* and *Orthorubulavirus*. (B) Homology of amino acid sequences of the L protein analyzed using WebLogo sequence logo generator. (C) Predicted model of P protein-bound SOSV L protein. RdRp, RNA-dependent RNA polymerase (colored in light blue); PRNTase, (Continued on next page)

Fig 5 (Continued)

polyribonucleotidyltransferase (colored in green); CD, connecting domain (colored in orange); MTase, methyltransferase (colored in yellow); CTD, C-terminal domain (colored in blue). In an enlarged view of the C-terminal loop, carbon atoms of which are colored in pink, residues from Asp2266 to Asp2270 are shown in stick diagrams. P protein tetramer bound to L protein are colored purple. (D) The SOSV minigenome-2 system was used to study the effect of amino acid substitution in the $D_{2265}LAGD_{2269}$ of SOSV L. Minigenome-2 and helper plasmids containing SOSV NP, P, and L (normal or mutant) were transfected to BHK/T7-9 cells. The relative Nluc expression when L normal is set to 1 is shown. Data represent means and standard deviations from three different experiments. L− indicates the values using an empty plasmid instead of SOSV L.

5′-CGnnnn-3′)—referred to as "phasing"—was critical for promoter activity of these *Rubulavirinae* (22, 23). However, our results using the SOSV minigenome-2 system demonstrated that even if the 5′-CG-3′ motif is not positioned at the first or second nucleotide of hexamers 13 through 16, the polymerase still exhibits a certain level of activity, provided that the nucleotide sequences of PE1 and PE2 are properly juxtaposed (Fig. 3B; front). Conversely, even when the 5′-CGnnnn-3′ sequence is correctly positioned within NP monomers in hexamers 13 through 16, promoter activity is lost if the alignment between PE1 and PE2 is disrupted (Fig. 3B; back). Thus, we found that even if the position of the 5′-CG-3′ motif within NP monomers is inaccurate, promoter function is maintained at a certain level of activity as long as the nucleotide sequences corresponding to [hex14 and hex1] and [hex15 and hex2]—e.g., hex14 (5′-CGcugg-3′) and hex1 (5′-cuuggu-3′)—are juxtaposed. In nature, when PE1 and PE2 sequences act as a promoter for genome replication, their nucleotide sequences are encapsidated by NP. Nevertheless, the juxtaposition of these two sequences is critical for promoter function. Further structural analysis is required to understand how the L protein recognizes the complex promoter structure formed by PE1 and PE2. Moreover, to understand the functions of PE1 and PE2, comprehensive studies are needed not only on SOSV but also on PIV5, hPIV2, and other viruses belonging to the genera *Orthorubulavirus* and *Pararubulavirus*.

In the *Rubulavirinae* subfamily, the agPE2 region contains three or four consecutive CGnnnn repeats (Fig. 4). To date, only three Rubulavirus species have been found to harbor four such repeats in their agPE2, and SOSV is the sole member of the *Pararubulavirus* genus (Fig. 4). Notably, our data showed that this fourth CG repeat is functional for minigenome promoter (Fig. 2), indicating that viruses bearing four repeats may gain a distinct advantage over those with only three.

A distinguishing feature of *Rubulavirinae* compared to *Orthoparamyxovirinae* and *Avulavirinae* is their different RNA editing mode (24). Paramyxoviruses utilize an RNA editing mechanism within the P/V gene, which encodes two proteins: the P protein, an essential polymerase cofactor for viral replication, and the V protein, an accessory protein that interacts with multiple host cellular proteins and acts as an interferon antagonist (25, 26). During mRNA transcription, the viral RdRp recognizes a cis-acting RNA editing signal within the P/V gene, and non-template-dependent nucleotides are co-transcriptionally inserted into the newly synthesized mRNA, allowing the synthesis of both P and V proteins from a single P/V gene (27–29). In many *Orthoparamyxovirinae* and *Avulavirinae*, the P protein is produced when RNA editing does not occur, while the V protein is generated when RNA editing occurs (referred to as "V-mode viruses"). In contrast, in viruses belonging to the subfamily *Rubulavirinae* (including MuV and Mapuera virus), the opposite occurs: V protein is produced when RNA editing does not occur, and P protein is synthesized when RNA editing occurs (known as "P-mode viruses"). Interestingly, P-mode viruses are also found outside the *Rubulavirinae* subfamily, such as Fer-de-lance virus (*Orthoparamyxovirinae*) (30), Anaconda paramyxovirus (*Orthoparamyxovirinae*) (31), and avian paramyxovirus 11 (*Avulavirinae*) (32). These viruses generate P proteins through RNA editing but belong to the "PE2 Out-ORF" type, where the agPE2 is located outside the L ORF, a feature they share with MuV and Mapuera virus. Therefore, the presence of agPE2 within the L ORF is not a strict requirement for a virus to adopt the P-mode. To date, no V-mode virus with an In-ORF type has been discovered, suggesting that if these two characteristics are indeed mutually exclusive, understanding the underlying

mechanism could shed light on why paramyxoviruses have evolved to adopt either P-mode or V-mode, as well as In-ORF or Out-ORF configurations.

As repeatedly mentioned, with the exception of MuV and Mapuera virus, *Rubulavirinae* possesses an agPE2 sequence within the L ORF, where the sequence "CGnnnn" is repeated at least three times (Fig. 5A) (15). This agPE2 sequence imposes strong codon selection constraints, leading to the semi-essential presence of the "D/G-I/L-G-B motif" at the C-terminus of the L protein (Fig. 5A and B). However, in the experiment using the SOSV minigenome-2 system, we observed that mutations in the corresponding "$D_{2265}LAGD_{2269}$ sequence" of this motif do not affect polymerase activity (Fig. 5D). While we cannot conclude that this motif is essential for viral replication, we cannot dismiss the possibility that it played some functional roles in viral evolution. From an evolutionary standpoint, it is likely that MuV and Mapuera virus evolved as PE2 In-ORF viruses, like many other viruses in the *Rubulavirinae*. Phylogenetic analysis of the L protein, as shown in Fig. 6A, suggests that MuV and Mapuera virus are not direct ancestors of *Rubulavirinae* and are unlikely to have emerged as a new independent lineage. Instead, it is more plausible that the PE2 In-ORF type virus appeared first in *Rubulavirinae*, and the PE2 Out-ORF type viruses, such as MuV and Mapuera virus, emerged later. Moreover, it is likely that these two viruses are not independent evolutionary lineages, but rather variants of PE2 In-ORF *Orthorubulavirus* genus that incidentally arose. The D/G-I/L-G-B motif likely played a critical role in this evolutionary process, which may explain why it remains conserved in MuV and Mapuera virus. Then, how did the L proteins of MuV and Mapuera virus arise? Their genomes and L amino acid sequences are not particularly long compared to other *Rubulavirinae* (Fig. 6B). Therefore, it seems more likely that the agPE2 was originally present within the L ORF of an ancestral virus (Fig. 6C; ancestral rubulaviruses). Over time, the C-terminus of the L protein in these viruses may have been lost, and despite the loss of the agPE2 sequences (Fig. 6C; truncation of L C-terminus), the virus could have evolved to reacquire the D/G-I/L-G-B motif (Fig. 6C; re-acquisition of D/G-I/L-G/B motif). This suggests that the D/G-I/L-G-B motif was crucial for *Rubulavirinae* and needed to be reacquired even after being lost. Because this motif is absent in the *Orthoparamyxovirinae* and *Avulavirinae* subfamilies, understanding its functional significance will be key to uncovering the unique features of the *Rubulavirinae* subfamily within the *Paramyxoviridae* family. In the structural model we constructed, the SOSV $D_{2265}LAGD_{2269}$ motif was found to be positioned in close proximity to a domain involved in RNA synthesis (e.g., the MTase domain), suggesting its involvement in gene expression (Fig. 5C). However, it is also possible that this motif is associated with other functions, such as virion assembly or interaction with host factors to improve the efficiency of viral propagation.

Currently, experiments using the SOSV minigenome-2 system (*Pararubulavirus* genus) have shown the importance of fourth CGnnnn for genome replication (Fig. 2) and that the D/G-I/L-G-B motif does not affect polymerase activity (Fig. 5D). However, comparative studies with other *Rubulavirinae*, such as *Orthorubulavirus* genus, may yield different results. Comprehensive comparative experiments would provide new insights into the differences between *Pararubulaviruses* and *Orthorubulaviruses* genera. Such research may further clarify the characteristics of SOSV, the only *Pararubulavirus* known to exhibit high pathogenicity in humans.

## MATERIALS AND METHODS

### Cells

Baby hamster kidney (BHK) cells constitutively expressing T7 RNA polymerase (BHK/T7-9 cells) (33) were cultured in Dulbecco's modified Eagle's medium with 5% fetal calf serum and penicillin/streptomycin. Cells were cultured at 37°C in 5% $CO_2$.

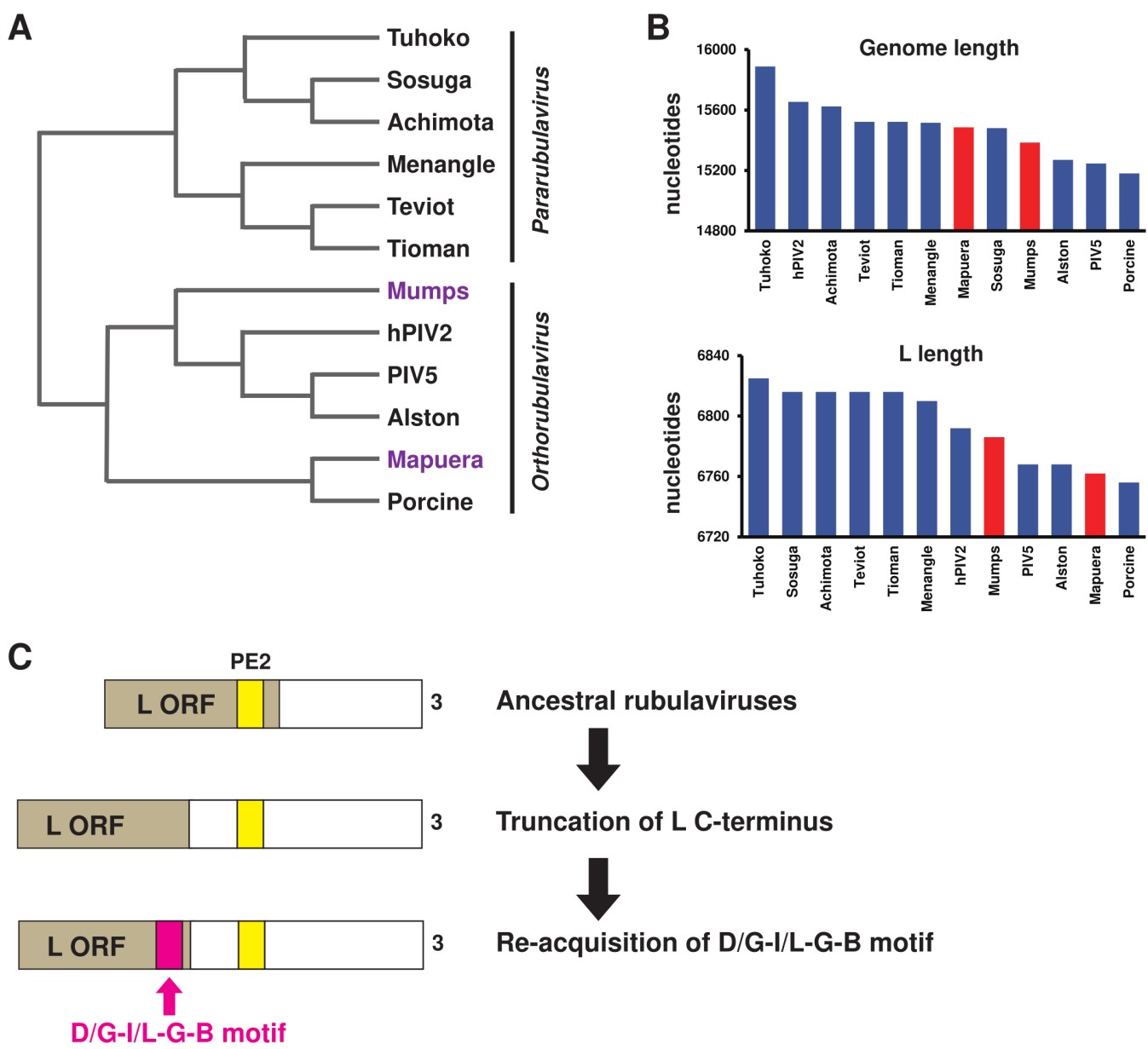

**FIG 6** Predicted evolutionary history of Rubulavirus. (A) Phylogenetic tree created from an alignment of the viral L protein of *Rubulavirinae*. (B) The length of genome nucleotides and L protein amino acids. The GenBank accession numbers used in this study are Mapuera (NC_009489.1), Porcine (NC_009640.1), mumps (NC_002200.1), PIV5 (KY685075.1), hPIV2 (AB176531.1), Alston (NC_055508.1), Menangle (AF326114.2), Sosuga (KF774436.1), Achimota (JX051319.1), Teviot (MH708896.1), Tioman (AF298895.2), and Tuhoko (GU128080.1). (C) Predicted evolutionary history of PE2 Out-ORF *Rubulavirinae*.

## Plasmid construction

The Nluc-expressing SOSV minigenome plasmid (SOSV-Nluc) was constructed using the pUC57 plasmid backbone. The Nluc gene was flanked by the 3′ UTR containing the leader sequence and the 5′ UTR containing the trailer sequence of the SOSV genome (GenBank accession number: NC_025343.1). The minigenome is set under the control of the T7 RNA polymerase promoter, and the transcript expressed as a negative sense RNA is cleaved at both ends by a hammerhead ribozyme and a hepatitis delta virus ribozyme (34). The SOSV NP, P, and L genes are set into the cloning site of the pCAGGS vector. All SOSV plasmids were generated by GenScript's custom gene synthesis services (GenScript Japan, Tokyo, Japan). A pCAGGS-derivative plasmid for the expression of firefly

luciferase (Fluc) was also constructed (15). The mutations for the agPE2 in SOSV-Nluc were introduced by a standard cloning method.

## SOSV Nluc minigenome assay

The SOSV Nluc minigenome assay was performed in BHK/T7-9 cells cultured in 12-well plates ($1 \times 10^5$ cells/well). Plasmids including SOSV-Nluc (0.5 µg), pCAGGS-L (0.4 µg), pCAGGS-P (0.4 µg), and pCAGGS-NP (0.4 µg) or an empty vector, and Fluc (0.05 µg) were transfected using XtremeGENE HP (Merck, Darmstadt, Germany). At 48 h post-transfection, the cells were lysed with passive lysis buffer (Promega), and the Nluc and Fluc activities were measured using the Nano-Glo Dual-Luciferase Reporter Assay System (Promega, Madison, WI, USA) according to the manufacturer's instructions. All activity values measured for Nluc were normalized to the expression levels of Fluc.

## Comparative analysis of *Rubulavirinae* PE2

Sequences of viruses in the subfamily *Rubulavirinae* were downloaded from the NCBI refseq database as previously reported (15). For each sequence, we extracted the sequences of the agPE2 (73–96 nts from the 3′ terminus of the antigenome). The sequence conservation was analyzed using the WebLogo sequence logo generator (https://weblogo.berkeley.edu/logo.cgi).

## AlphaFold predictions

The AlphaFold models of SOSV L protein were predicted using the AlphaFold version 2.0 algorithm on the Colab server (https://colab.research.google.com/github/sokrypton/ColabFold/blob/main/AlphaFold2.ipynb), accessed on 25 July 2024 (35). Predictions were performed with default multiple sequence alignment generation using the MMSeqs2 server, with 48 recycles and templates (homologous structures). The complex structure of L protein and P protein tetramer was generated based on the best out of L protein AlphaFold models (rank 1) and the structure of mumps virus L protein bound by P protein tetramer (PDB code 8IZL). The figures were drawn using PyMOL software (36).

## Phylogenetic tree of *Rubulavirinae*

Alignment and phylogenetic reconstructions were performed using the function "build" of ETE3 3.1.3 (37) as implemented on the GenomeNet (https://www.genome.jp/tools/ete/). ML tree was inferred using PhyML v20160115 ran with model and parameters: --pinv e --alpha e --nclasses 4 -o tlr -f m --bootstrap -2 (38). Branch supports are the $\chi^2$-based parametric values returned by the approximate likelihood ratio test.

## Statistical analysis

A one-way analysis of variance (ANOVA) was performed, followed by Tukey's honest significant difference (HSD) test for *post hoc* pairwise comparisons. The ANOVA was conducted using the aov() function, and the subsequent Tukey's HSD test was performed with the TukeyHSD() function in R (version 4.4.0, R Foundation for Statistical Computing, Vienna, Austria). Statistical significance was assigned when $P$ values were <0.05.

## ACKNOWLEDGMENTS

We thank Dr. Naoto Ito (Gifu University) for providing the BHK/T7-9 cells. We also thank Dr. Kyoko Tsukiyama-Kohara (Kagoshima University) and members of her laboratory for their support with the biological experiments.

   This work was supported by grants from the Japan Agency for Medical Research and Development (AMED) Research Program on Emerging and Re-emerging Infectious Diseases JP23fk0108687 (to Y.M.), the JSPS KAKENHI grant number 24K09229 (to Y.M.), the Takeda Science Foundation (to Y.M.), the Kato Memorial Bioscience Foundation (to Y.M.), and the Kieikai Research Foundation (to Y.M.). This work was supported by the

Cooperative Research Program of the Institute for Life and Medical Sciences, Kyoto University, and the Grant for International Joint Research Project of the Institute of Medical Science, the University of Tokyo.

## AUTHOR AFFILIATIONS

[1]Transboundary Animal Diseases Research Center, Joint Faculty of Veterinary Medicine, Kagoshima University, Kagoshima, Kagoshima, Japan

[2]Graduate School of Medicine, Chiba University, Chiba, Chiba, Japan

[3]Department of Pathology and Parasitology, Faculty of Veterinary Medicine, Chattogram Veterinary and Animal Sciences University, Chattogram, Bangladesh

[4]Department of Virology 3, National Institute of Infectious Diseases, Shinjuku, Tokyo, Japan

[5]Center of Quality Management Systems, National Institute of Infectious Diseases, Shinjuku, Tokyo, Japan

[6]Genome Immunobiology RIKEN Hakubi Research Team, RIKEN Center for Integrative Medical Sciences, Yokohama, Kanagawa, Japan

[7]Department of Pharmaceutical Chemistry, Faculty of Pharmacy, Yasuda Women's University, Hiroshima, Japan

## AUTHOR ORCIDs

Junna Kawasaki http://orcid.org/0000-0002-6609-5300
Tofazzal Md. Rakib http://orcid.org/0000-0003-2642-9908
Yusuke Matsumoto http://orcid.org/0000-0003-3500-2279

## FUNDING

| Funder | Grant(s) | Author(s) |
|---|---|---|
| Japan Agency for Medical Research and Development (AMED) | JP23fk0108687 | Yusuke Matsumoto |
| MEXT | Japan Society for the Promotion of Science (JSPS) | 24K09229 | Yusuke Matsumoto |
| Takeda Science Foundation | none | Yusuke Matsumoto |
| Kato Memorial Bioscience Foundation | none | Yusuke Matsumoto |
| Kieikai Research Foundation | none | Yusuke Matsumoto |

## AUTHOR CONTRIBUTIONS

Lipi Akter, Data curation, Formal analysis, Writing – review and editing | Junna Kawasaki, Formal analysis, Investigation, Writing – review and editing | Tofazzal Md. Rakib, Data curation, Writing – review and editing | Takashi Okura, Formal analysis, Investigation, Writing – review and editing | Fumihiro Kato, Data curation, Formal analysis, Writing – review and editing | Shohei Kojima, Data curation, Formal analysis, Writing – review and editing | Kosuke Oda, Data curation, Formal analysis, Visualization, Writing – review and editing | Yusuke Matsumoto, Conceptualization, Data curation, Formal analysis, Funding acquisition, Investigation, Methodology, Project administration, Resources, Supervision, Validation, Writing – original draft, Writing – review and editing

## DATA AVAILABILITY

All databases used in this study are available from DDBJ/ENA/GenBank (https://www.ddbj.nig.ac.jp/about/insdc-e.html). The accession numbers of the viral sequences used in this study are listed in Table S1.

## ADDITIONAL FILES

The following material is available online.

### Supplemental Material

**Table S1 (Spectrum00534-25-s0001.xlsx).** Viral genome sequence list.

### Open Peer Review

**PEER REVIEW HISTORY (review-history.pdf).** An accounting of the reviewer comments and feedback.

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
