## [Reviewer comments · Microbiology Spectrum]

Microbiology Spectrum

Functional analysis of promoter element 2 within the viral polymerase gene of an emerging paramyxovirus, Sosuga virus

Lipi Akter, Junna Kawasaki, Tofazzal Md Rakib, Takashi Okura, Fumihiro Kato, Shohei Kojima, Kosuke Oda, and Yusuke Matsumoto

Corresponding Author(s): Yusuke Matsumoto, Kagoshima University

Review Timeline:

Submission Date:	February 24, 2025
Editorial Decision:	February 27, 2025
Revision Received:	March 10, 2025
Accepted:	March 12, 2025

Editor: Takamasa Ueno

Reviewer(s): The reviewers have opted to remain anonymous.

Transaction Report:

DOI: <https://doi.org/10.1128/spectrum.00534-25>

Re: Spectrum00534-25 (Functional analysis of promoter element 2 within the viral polymerase gene of an emerging paramyxovirus, Sosuga virus)

Dear Dr. Yusuke Matsumoto:

I am pleased to inform you that your manuscript has been editorially accepted for publication. However, there are a few additional questions in the submission form that need to be answered before the final decision. Once these are completed, please return your submission so that I can move your paper forward to acceptance.

Sincerely,
Takamasa Ueno
Editor
Microbiology Spectrum

Re: Spectrum00534-25R1 (Functional analysis of promoter element 2 within the viral polymerase gene of an emerging paramyxovirus, Sosuga virus)

Dear Dr. Yusuke Matsumoto:

Your manuscript has been accepted, and I am forwarding it to the ASM production staff for publication. Your paper will first be checked to make sure all elements meet the technical requirements. ASM staff will contact you if anything needs to be revised before copyediting and production can begin. Otherwise, you will be notified when your proofs are ready to be viewed.

Sincerely,
Takamasa Ueno
Editor
Microbiology Spectrum